# NT-proBNP—Possible Prognostic Marker in Pregnant Patients with Associated Cardiovascular Risk Factors and SARS-CoV-2 Infection

**DOI:** 10.3390/diagnostics13193032

**Published:** 2023-09-23

**Authors:** Carmen-Ioana Marta, Anca Laura Maghiari, Elena Bernad, Lavinia Stelea, Brenda Bernad, Lioara Boscu, Radu Neamtu, Adrian Gluhovschi, Mircea Diaconu, Catalin Dumitru, Bogdan Sorop, Katalin Babes

**Affiliations:** 1Faculty of Medicine and Pharmacy of Oradea, University of Oradea, 410087 Oradea, Romania; silaghi.carmen@gmail.com (C.-I.M.); piszekati@yahoo.co.uk (K.B.); 2Clinic of Obstetrics and Gynecology, “Pius Brinzeu” County Clinical Emergency Hospital, 300723 Timisoara, Romania; bernad.elena@umft.ro (E.B.); stelea.lavinia@umft.ro (L.S.); neamtu.radu@umft.ro (R.N.); gluhovschi.adrian@umft.ro (A.G.); diaconu.mircea@umft.ro (M.D.); dumitru.catalin@umft.ro (C.D.); bogdan.sorop@umft.ro (B.S.); 3Department I—Discipline of Anatomy and Embryology, “Victor Babes” University of Medicine and Pharmacy, 300041 Timisoara, Romania; 4Department of Obstetrics and Gynecology, Faculty of Medicine, “Victor Babes” University of Medicine and Pharmacy, 300041 Timisoara, Romania; 5Center for Neuropsychology and Behavioral Medicine, “Victor Babes” University of Medicine and Pharmacy, 300041 Timisoara, Romania; 6Doctoral School, “Victor Babes” University of Medicine and Pharmacy, 300041 Timisoara, Romania; bernad.brenda@umft.ro (B.B.); lioara.boscu@umft.ro (L.B.); 7Clinical County Emergency Hospital of Oradea, 410167 Oradea, Romania

**Keywords:** NT-proBNP, SARS-CoV-2, cardiovascular diseases, pregnancy complications, peripartum cardiomiopathy

## Abstract

Background: N-terminal pro brain-type natriuretic peptide (NT-proBNP) is a practical biomarker in the clinical pathologies where the ventricle is under stress and particularly stretched in the general population. The study aims to compare the value of NT-proBNP and its importance in the prognosis and severity of the cases involving pregnant patients with SARS-CoV-2 infection and cardiovascular risk factors to those of low-risk pregnant patients, mainly by analysing their symptoms, administered medication, days of hospitalization and severity of the viral disease. Methods: The study included a total of eighty-three pregnant patients who underwent natural birth or caesarean section at out hospital. NT-proBNP levels were analyzed at hospital admission as a potential cardiovascular marker. A comparative analysis was performed between pregnant patients with cardiovascular risk factors and pregnant patients without cardiovascular risk factors regarding NT-proBNP values. Results: Pregnant patients with SARS-CoV-2 infection and cardiovascular risk factors had higher values of NT-proBNP in comparison to pregnant patients without cardiovascular risk factors. Conclusions: NT-proBNP testing in pregnant patients with SARS-CoV-2 infection can be a relatively important marker to be taken into consideration when it comes to the management, treatment and outcome of the cases, especially when it comes to women with associated cardiovascular risk factors.

## 1. Introduction

N-terminal pro brain-type natriuretic peptide (NT-proBNP) is a practical biomarker in the clinical pathologies where the ventricle is under stress and particularly stretched in the general population [1]. Hemodynamic modifications happening while pregnant can expand the stress on the mother’s cardiovascular system in patients that already have a cardiac disease or reveal cardiovascular pathology in formerly healthy subjects [2]. The diagnostic value of biomarkers in pregnancy-associated pathologies associated with cardiac stress, such as preeclampsia, gestational hypertension and gestational diabetes, are not well defined [3]. NT-proBNP is often measured when it comes to diagnosing heart failure in pregnant patients. Some studies also indicate that high NT-proBNP values in early pregnancy can be correlated with the risk of developing adverse pregnancy outcomes, such as preeclampsia [4]. NT-proBNP has a major prognostic role in patients diagnosed with peripartum cardiomyopathy and is notably elevated in women with acute peripartum cardiomyopathy in comparison with healthy postpartum patients [5]. Even though COVID-19 has been related to a lot of cardiovascular pathologies (such as venous thromboembolic disease, acute arterial thrombotic events and arrhythmias), the correlation of these clinical events with high values of natriuretic peptide is not well defined [6]. Presently, there are no accepted reference values for NT-proBNP in pregnant patients, especially for those with SARS-CoV-2 infection, making it difficult to assess the NT-proBNP high values and its severity prognosis [7]. Therefore, our study conducts a comparative analysis in pregnant patients with a SARS-CoV-2 infection and that have associated cardiovascular risk factors or not. The viral infection with SARS-CoV-2 (omicron variant) associated with pregnancy has been shown to be linked to an increased risk of severe complications and maternal morbidity, especially for the patients that were symptomatic and unvaccinated. Pregnant patients with severe symptoms of COVID-19 were associated with an increased risk of pre-eclampsia [8]. Our assumption is that pregnant women with SARS-CoV-2 infection and associated cardiovascular risk factors will have higher values of NT-proBNP and an unfavorable prognosis when it comes to the recovery from the viral infection in comparison with pregnant women with SARS-CoV-2 infection but no cardiovascular risk factors. The study aims to compare the value of NT-proBNP, which is a marker frequently associated with cardiac stress, and its importance in the prognosis and severity of the cases involving pregnant patients with SARS-CoV-2 infection and cardiovascular risk factors to those of low-risk pregnant patients is mainly performed by analysing their symptoms, administered medication, days of hospitalization and severity of the viral disease. The results of this comparative analysis highlights the potential impact of the NT-proBNP value in SARS-CoV-2 infection, especially when it comes to pregnant patients with cardiovascular risk factors and their prognosis. In addition, our findings are of importance to contributing in clinical practice and managing pregnant patients with SARS-CoV-2 viral infection and cardiovascular risk factors, in order to improve the case evolution and outcome. 

## 2. Materials and Methods

The study was conducted from 1 January 2021 to 31 December 2022 at the “Pius Brinzeu” County Emergency Clinical Hospital from Timisoara in the Department of Obstetrics and Gynecology in association with “Victor Babes” University of Medicine and Pharmacy from Timisoara. Medical records of our patients were obtained with patients’ consent and stored in a database that respected privacy laws. The collected data contained information such as: patient medical history, demographic data and medical or surgical procedures. 

Ethical approval (No. 06/15 January 2021) was obtained for this study from “Pius Brinzeu” County Emergency Clinical Hospital from Timisoara management. This approval document ensures the protection of the participants’ personal medical data, rights and well-being through our study. In addition, this was a devotion to conduct the research in an ethical approach.

The patients that participated in our study received explanations regarding the study’s purpose, which was to elucidate any fluctuations in NT-proBNP levels during pregnancy and the potential modifications to these levels caused by SARS-CoV-2 infection, information present in their informed consent. Participants were to be informed about the procedures required, such as blood drawing, and were aware of any minor complications, such as brief soreness at the puncture site. Their autonomy, confidentiality and ability to withdraw at any time without risking their standard care were all protected.

The study included a total of 83 pregnant patients who underwent natural birth or caesarean section at our hospital. Caesarean section is a surgical intervention in which the baby is delivered through an incision in the mother’s pelvis and uterus. It is usually performed when vaginal birth is not an option and the mother or the baby are at risk if they undergo a natural birth. 

Nasopharyngeal and oropharyngeal swabbing was used to acquire samples for the analysis of the SARS-CoV-2 virus infection. A medical professional reached into the patient’s nostril, to the nasopharynx, and twisted a specific swab many times to guarantee adequate collection of epithelial cells while wearing the proper protective equipment against the viral infection. Through the mouth, this process was repeated in the oropharynx. In order to maintain the viability of the possible virus particles throughout transport to the testing facility, the swab was sealed in a sterile viral transport medium.

Reverse transcription polymerase chain reaction (RT-PCR) was performed to detect and quantify the SARS-CoV-2 virus in the obtained samples. Using a reverse transcription enzyme, the viral RNA was first transformed into complementary DNA (cDNA). Then, using a thermostable polymerase, certain sections of this DNA were amplified exponentially. The amplification process was monitored in real-time, and because special probes and primers were used, the viral target could be detected and quantified precisely. The virus’s presence was validated by identifying specific fluorescent signals emitted throughout the amplification process.

Venous blood samples were rigorously collected using a standard venipuncture technique to measure the concentration of NT-proBNP in patients. After ensuring the patient’s comfort and placing a tourniquet, a vein, often in the antecubital fossa, was located. After inserting a sterile needle into the vein, blood was extracted into a syringe or collection tube. The blood samples were collected and brought to the laboratory where they were centrifuged to separate the plasma from the blood cells. The presence of NT-proBNP in the plasma was then evaluated using an immunological technique known as solid-phase enzyme-linked immunosorbent assay (ELISA). This test employs particular antibodies that bind to NT-proBNP, allowing for the precise detection and quantification of this biomarker.

Our collected data was analyzed with GraphPad Prism (version 5). Before analyzing our collected data, we went through normality, missing values and outliners. If required, we normalized the variables to meet the presumption of the statistical tests. When balancing the means of the two studied groups, the *t*-test concluded if the difference was statistically important or the result was a random coincidence.

We used descriptive statistics to give an overview of the collected data, and group differences were analyzed using *t*-tests. All of our statistical tests conducted were two-tailed, and we took into consideration and gave statistical importance to *p*-values lower than 0.05. We reported the results as mean ± standard deviation. This is a statistical measure that indicates the size of variability or dispersion in a dataset. GraphPad Prism was used to generate the tables and figures in order to guarantee an explicit presentation of the results.

We also used the z-test for binomial proportions to differentiate between the two cohorts’ observed percentages. This statistical method is appropriate for comparing proportions made from two distinct groups, especially when dealing with categorical data, as demonstrated in the demographic information of our cohorts. The purpose of this test is to determine whether or not there is a statistically significant difference in the observed proportions of the two groups. The analytical process involves calculating a z-statistic, which is then compared to a typical normal distribution to determine the appropriate *p*-value. A small *p*-value indicates a more noticeable difference between the proportions, offering insight into whether the observed variance is likely due to random chance or represents a true difference between the cohorts.

Inclusion criteria for the study:-All of the patients were pregnant in the second and third trimester;-Pregnant patients with SARS-CoV-2 infection (two positive RT-PCR consecutive tests in our institution);-Pregnant patients over the age of 18;-Women who underwent regular check-ups and monitored the pregnancy;-Patients who had given their informed consent for the study;-Patients with mild, moderate of severe symptoms of SARS-CoV-2 infection;-Asymptomatic pregnant patients who tested positive for SARS-CoV-2 infection;-Patients who have received prenatal care.-We excluded the patients who met the following criteria:-Women who are incapable of giving consent for studies;-Patients with a previous history of substance abuse;-Patients with known psychiatric disorders;-Women with pre-existing renal dysfunction;-Patients who had been involved in other clinical trials or studies in the last twelve months;-Pregnant patients with inadequately managed endocrine or metabolic disorders.

In the group of patients with cardiovascular risk, we included pregnant patients who met at least one of the following criteria:-Pregnancy-induced hypertension;-Preexisting hypertension;-Preeclampsia;-Eclampsia;-Type 1 and Type 2 diabetes;-Leading predominant sedentary lifestyle;-Smoking history of at least 5 years;-Triglycerides over 200 mg/dL;-Family history of cardiovascular disease;-Total cholesterol over 280 mg/dL;-Unhealthy dietary habits such as excessive consumption of added sugars, sodium, saturated and trans fats.

## 3. Results

The participants that were included in our study and have met all the requested criteria were divided in two groups: Group 1, which consisted of 46 pregnant patients that had no cardiovascular risk or disease, and Group 2, which included 37 pregnant patients with cardiovascular risk or disease.

As shown above, in Table 1, we present a comparison between our two groups, referred to as Group 1 and Group 2, based on the value of NT-proBNP that was measured in pg/mL. The table offers statistical measurements and results for each of our studied group, indicating differences between the two groups. The table proves that Group 1 and Group 2 have different distributions of data. Group 2, consisting of patients with cardiovascular risk, has higher values across all measured statistics compared to Group 1 that does not have a cardiovascular rick. The small *p*-value suggests that there is an important difference between the two studied groups, likely related to the parameter being measured. 

In a comparative analysis of associated pathologies between Group 1 (comprising 46 patients) and Group 2 (consisting of 37 patients), several noteworthy observations can be made based on *p*-values. For hypothyroidism, 8.69% of Group 1 patients presented with this condition, in contrast to 13.51% in Group 2 with a *p*-value of 0.4633, indicating no statistically significant difference between the two groups. A similar tendency is seen for obstructive sleep apnea with a *p*-value of 0.823. However, the difference in the incidence of thrombophilia is noticeable with a *p*-value of <0.001, denoting a highly statistically significant difference. For bronchial asthma, we have a *p*-value of 0.074, which borders on statistical significance. Chronic viral infections manifested in 21.73% of Group 1 patients as opposed to 8.10% in Group 2, giving a *p*-value of 0.072, similarly nearing significance. Lastly, the occurrence of autoimmune diseases was 17.39% in Group 1 and 10.81% in Group 2 with a *p*-value of 0.369, suggesting no significant difference in this pathology between the two groups.

The patients included in our study presented associated pathologies, which did not influence the value of NT-proBNP, as opposed to their biological status and evolution. Distribution of the associated pathologies is presented below in Table 2. The *p*-values associated with each pathology provide valuable insights into the statistical significance of observed differences. For hypothyroidism, we have a *p*-value of 0.4633. This *p*-value suggests that there is no statistically significant difference in the occurrence of hypothyroidism between the two groups. Similarly, in the case of obstructive sleep apnea, we found a *p*-value of 0.823, indicating no significant distinction. However, the most striking contrast is observed in thrombophilia, where difference deliver a highly significant *p*-value of <0.001. For bronchial asthma, chronic viral infections and autoimmune diseases, the *p*-values are 0.074, 0.072 and 0.369, respectively, signifying varying degrees of significance or lack thereof for these pathologies between the two groups.

As for medical characteristics and outcomes, Table 3 clearly demonstrates a substantial difference between the two analyzed groups when it comes to pregnant patients with cardiovascular risk or associated diseases. Group 2 had more pathological findings and required more medical resources as shown above. A great majority of our pregnant patients with cardiovascular associated risks gave birth through cesarean section because of the symptomatology that could influence in a negative way the vaginal birth outcome of the mother and the newborn. Notably, for patients living urban areas, there’s a significant difference in prevalence offering a *p*-value of 0.024. The same trend is observed for patients from rural areas with a *p*-value of 0.0245. Cesarean section deliveries are significantly more frequent in Group 2 (91.89%) compared to Group 1 (34.78%), resulting in a *p*-value of <0.001. Conversely, normal deliveries are significantly more common in Group 1 (65.21%) than in Group 2 (8.10%) with a *p*-value of <0.001. Smoking patients are significantly more prevalent in Group 2 (70.27%), reflected in a *p*-value of <0.001, while non-smoking patients are more common in Group 1 (100%). Additionally, several medical treatments and outcomes, such as pathological ECG findings, oxigenotherapy, hospitalization durations exceeding 10 days and mortality rates, show significant differences between the two groups, as highlighted by *p*-values of <0.001. However, other factors, such as AIRVO usage, tracheal intubation and certain hospitalization durations, do not exhibit statistically significant differences. These *p*-values collectively enlighten the varying clinical characteristics and outcomes between the two groups that are being studied.

Table 4 provides information on various pathological findings observed in electrocardiograms (ECG) along with their respective frequencies. Ventricular extrasystoles were identified in 3 cases (3.61%). Ventricular extrasystoles refer to abnormal heartbeats that originate from the ventricles, causing an irregular rhythm.

Sinus bradycardia was observed in 4 cases (4.81%). Sinus bradycardia is a condition characterized by a slower-than-normal heart rate originating from the sinus node, which can result in reduced cardiac output.

Two cases (2.4%) experienced sinus tachycardia. Sinus tachycardia refers to a consistent heart rhythm with a faster-than-normal heart rate, leading to an elevated cardiac output.

Similarly, two cases (2.4%) presented with long QT syndrome. The QT interval on an ECG represents the duration of the ventricular action potential, reflecting the physiological duration of ventricular depolarization and repolarization.

Two cases (2.4%) demonstrated larger T waves. Tall T waves can be an indicator of ischemic changes or hyperkalemia or may also occur as a normal variation, depending on their distribution in the precordial leads.

The presence of these pathological findings in the electrocardiograms highlights the need for further evaluation and management to address any underlying cardiac abnormalities and ensure appropriate patient care.

In Table 5, we presented the frequency of SARS-CoV-2 symptoms amongst our studied patients at admission and the difference between the two groups. As it is shown below, in the second group, with cardiovascular risk, symptoms were more frequent and more severe. The calculated *p*-values highlight the statistical significance of differences in reported symptoms. Notably, there is no significant difference in the prevalence of ageusia between the two groups with *p*-value 0.164. Similarly, anosmia shows no significant variation with a *p*-value of 0.124. Cough, a common COVID-19 symptom, is reported in 86.95% of Group 1 and 94.59% of Group 2, but the difference is not statistically significant (*p*-value 0.214). In contrast, fever shows a highly significant difference with 95.65% of patients in Group 1 reporting fever compared to 51.35% in Group 2 with a *p*-value of <0.001. Dyspnea, rhinorrhea and fatigability do not exhibit statistically significant differences between the two groups with *p*-values of 0.246 while the presence of patients remaining asymptomatic upon admission is significantly different with *p*-value 0.0262. These *p*-values provide valuable insights into the variation in COVID-19 symptomatology between the two groups.

The patients divided in the two groups received the following medication while admitted in our hospital: low molecular mass anticoagulant, corticoid therapy, vitamins, antibiotics, antalgics, non-steroidal anti-inflammatory drugs and antivirals distributed as shown below in Table 6. The *p*-values associated with each medication category highlights the statistical significance of differences in medication usage between the two groups. Notably, the use of low molecular mass anticoagulants differs significantly with 78.26% in Group 1 compared to 100% in Group 2, resulting a *p*-value of 0.0013. Corticoid therapy shows a highly significant difference with only 32.60% in Group 1 and 89.18% in Group 2 with a *p*-value of <0.001. The usage of vitamins also shows a statistically significant difference, as all patients in Group 1 are administered vitamins compared to 94.59% in Group 2 with a *p*-value of 0.0475. Similarly, antibiotic usage is significantly different with 58.69% in Group 1 and 78.37% in Group 2, resulting in a *p*-value of 0.0434. The use of antalgics and non-steroidal anti-inflammatory drugs (NSAIDs) demonstrate statistically significant differences between the groups with *p*-values of 0.0562 and <0.001, respectively. Additionally, the administration of antiviral medications significantly differs with 19.56% in Group 1 compared to 81.08% in Group 2, resulting in a *p*-value of <0.001. These *p*-values collectively highlight variations in medication usage patterns between the two groups, providing valuable insights into treatment approaches for SARS-CoV-2 infection.

Figure 1 demonstrates the difference of the standard deviation between Group 1 and Group 2 regarding the NT-proBNP value.

## 4. Discussion

The comparative analysis of NT-proBNP levels in pregnant patients diagnosed with SARS-CoV-2 infection with cardiovascular risk versus low-risk pregnant patients gives us important understanding of the outcome, evolution and management of the cases. The study mainly evaluated and compared the NT-proBNP values and their influence and corroboration between the two groups.

NT-proBNP measurements must be interpreted as a continuous variable with “normal” values, such as 70 pg/mL, in order to exclude the diagnosis of heart failure [14].

The clinical use of NT-proBNP has not yet been studied enough in pregnant patients. Pregnancy can be a physiological stress for the cardiovascular system because of the 45–50% increase in intravascular volume [15].

The values of NT-proBNP are usually higher in pregnant patients in comparison with non-pregnant patients [16]. The diagnostic roles of NT-proBNP measurement in pregnant patients can involve: evaluating patients with symptoms of heart failure, evaluation of hypertensive pathologies in pregnancy, monitoring of pregnant patients with known cardiac disease and screening for left ventricular dysfunction [17]. The use of NP testing (including NT-proBNP) in pregnant patients with known cardiac pathology is of great interest and potential [18]. Some studies show a connection of NT-proBNP course with total peripheral resistance. Some indicate that peripheral resistance in physiological pregnancies is higher in the first trimester of pregnancy and decreases advancing in pregnancy towards 20–24 gestational weeks, followed by a small increase in the last weeks of gestation [19]. Furthermore, some studies evaluated the value of NT-proBNP in an emergency diagnosis of pulmonary embolism in pregnant patients, which also requires a reliable cut-off value [20]. Cardiac failure in pregnancy can have two contexts: heart failure in pregnant women with a known cardiac pathology or development of heart failure without a known cardiac pathology, such as peripartum cardiomyopathy [21]. 

NT-proBNP has a very important prognostic role in patients with peripartum cardiomyopathy [5]. Peripartum cardiomyopathy is likely to lead to systolic dysfunction, especially in black women. Pregnancy-related hypertension pathologies are the main hypertensive disorder of pregnant patients’ risk factor for peripartum cardiomyopathy. In addition to this, peripartum cardiomyopathy often occurs earlier in patients with positive hypertensive disorders of pregnancy results than in those with negative hypertensive disorders of pregnancy results. Hypertensive disorders of pregnancy that raise the risk of peripartum cardiomyopathy can comprise hypertension and preeclampsia, which are more often found in black women. Preeclampsia and peripartum cardiomyopathy are pathophysiological-related illnesses that are probably caused by increased placental production of vasculotoxic and antiangiogenic hormones. The earlier we can diagnose it, the better the outcome [22]. NT-proBNP is higher in women with acute peripartum cardiomyopathy in comparison to healthy postpartum patients. The NICE guideline suggest performing a transthoracic Doppler 2D echocardiography and specialist assessment for patients with suspected heart failure and NT-proBNP C400 pg/mL [23]. NT-proBNP has often high values in patients with SARS-CoV-2 and is greatly associated with myocardial injury and mortality [24]. We have multiple pathophysiological pathways that can be accountable for the high values of NT-proBNP levels after COVID-19 infection. Inflammation can be a possible factor for higher circulating natriuretic peptides [25]. It has been assumed that the high value of natriuretic peptides can be associated with weakening of cardiac function in SARS-CoV-2 [26]. In addition, pre-eclampsia was strongly associated in nulliparous patients with COVID-19 infection with no associated risk factors and no association with viral infection severity and form of disease [27]. Myocardial injury in SARS-CoV-2 has been the subject of a large-scale research. It may probably result in increased wall stress and consequently higher NT-proBNP values. Studies focused on magnetic resonance imaging have suggested an elevated prevalence of myocardial inflammation among these patients [28]. A multinational study performed on a large cohort of patients showed that pathologies such as diabetes mellitus and insulin-dependent gestational diabetes, being overweight and obese were risk factors for the viral infection in pregnancy [29]. A recent study proved that, as for the heart failure biochemical marker, N terminal pro B type natriuretic peptide (NT-proBNP) had high values during the course of hospitalization in the patients who passed away [30]. The mechanism of COVID-19 induced cardiac injury was still imprecise. As a result from the autopsy by Xu and colleagues, a few interstitial mononuclear inflammatory infiltrates were noticed in the heart biopsy, which indicated an inflammation induced by cardiac injury [31]. Values of >125 pg/mL in the algorithm for heart failure are generally considered high levels of NT-proBNP in patients [32]. The signal for a stronger association of high values of NT-proBNP with mortality in women is alarming and has to be evaluated in more prospective studies [33]. NT-proBNP > 128 pg/mL is currently used at 20 weeks of gestation as a predictive marker of event later in pregnancy [34]. Further research is needed when it comes to the importance of NT-proBNP testing as a severity prognosis marker in pregnant patients with SARS-CoV-2 infection, especially for those with cardiovascular risk factors. Another important aspect when it comes to pregnant patients and associated COVID-19 infection is the vaccination status against the viral infection and its influence on complications and the disease. It has been shown that despite the fact that vaccination did not have a strong impact on preventing the disease, positive outcomes, such as milder symptoms and fewer complications, were noticed. For a better outcome of the COVID-19 form of the disease, it is required to complete a full vaccination scheme and also administer the booster [8]. In addition to this, long-term monitoring of these patients is necessary and of high importance.

### Strengths and Limitations

One of the major strengths of this study is the fact that it is addressed to two separate groups of pregnant patients based on cardiovascular risk status and the viral infection with SARS-CoV-2. It is considered to be a relatively new subject that has not yet been addressed and studied sufficiently. The study tries to bring a new approach on the use of monitoring NT-proBNP when it comes to pregnant patients infected with SARS-CoV-2 that also had cardiovascular risk factors during this crisis period that involved a lot of uncertainty about treatment, management and evolution while encountering a new pathology that needed immediate attention. Another strength is that we examine the importance of the NT-proBNP higher value influence in the outcome of the way of giving birth and how this outcome can be influenced. Collecting data, such as medical history and demographic information, enables a detailed analysis of potential confounding factors. The study’s validity is strengthened by appropriate statistical analysis methods, such as *t*-tests. 

Despite its contributions, through our research, we found certain limitations. First of all, our data and tests that we analyzed in the study group were collected in one medical institution that can limit the diversity of the patient population. Another limitation that we had was the relatively small sample size and short period of time because of limitations during the pandemic period. In addition, we had no long-term follow-ups of our study groups because of the short period of time the study was conducted in. In addition to this, advanced research needs to be performed to see the implications and influence of COVID-19 in pregnant patients associated with pathologies such as diabetes mellitus, pre-eclampsia, intrauterine fetal growth restriction, gestational diabetes, preterm birth, still birth, perinatal death and maternal morbidity and mortality. Furthermore, we could not monitor the influence of the vaccination against COVID-19 viral infection and the severity of the disease, symptoms and other complications because most of the patients from the studied cohort were unvaccinated. Our goal is to continue to advance in this line of research within our institution while also associating with other prestigious institutions and researchers in this field. We are sure that expanding the limits of our research is critical to validate and corroborate our preliminary findings.

## 5. Conclusions

The comparative analysis of NT-proBNP levels in pregnant patients diagnosed with SARS-CoV-2 infection with cardiovascular risk versus low-risk pregnant patients gives us important implications in general patient care.

Higher values of NT-proBNP were found in pregnant patients that went through the most severe forms of SARS-CoV-2 infection admitted in our hospital, and unfortunately, some of them passed away. 

The pregnant patients who had the most severe forms of the infection and the worst outcome were included in the cardiovascular associated risk factors group.

The findings of our rigorous research are critical in prenatal care because they provide insights aimed at clarifying and enhancing cardio-vascular health measures for both the expecting woman and the fetus. However, it is critical to recognize the intricacies underlying the effects of NT-proBNP levels on the cardiovascular system.

In conclusion, we consider that NT-proBNP testing in pregnant patients with SARS-CoV-2 infection can be a relatively important marker to be taken into consideration when it comes to the management, treatment and outcome of the cases, especially when it comes to women with associated cardiovascular risk factors.

## Figures and Tables

**Figure 1 diagnostics-13-03032-f001:**
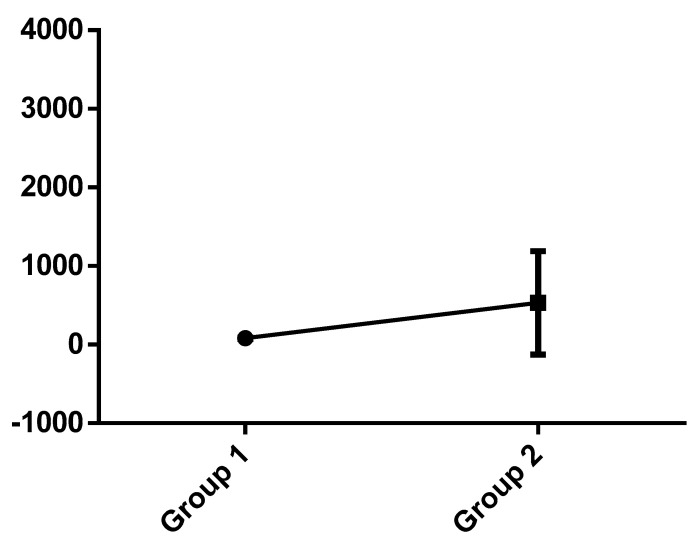
Standard deviation between groups.

**Table 1 diagnostics-13-03032-t001:** Descriptive statistics and significant differences between Group 1 and Group 2.

	Group 1 (*n* = 46)	Group 2 (*n* = 37)
Minimum	68.00	70.00
25% Percentile	73.00	97.50
Median	80.50	188.0
75% Percentile	87.00	943.0
Maximum	99.00	3238
Mean	81.07	531.4
Std. Deviation	9.222	659.0
*p*-value *t*-test	<0.0001	

**Table 2 diagnostics-13-03032-t002:** Comparative analysis of pathological conditions in Group 1 and Group 2.

Associated Pathologies	Group 1 (*n* = 46)	Group 2 (*n* = 37)	*p* Value
Hypothyroidism	4 (8.69%)	5 (13.51%)	0.4633
Obstructive Sleep Apnea	3 (6.52%)	2(5.40%)	0.823
Thrombophilia	2 (4.34%)	13 (35.13%)	<0.001
Bronchial asthma	6 (13.04%)	1 (2.70%)	0.074
Chronic viral infections	10 (21.73%)	3 (8.10%)	0.072
Autoimmune diseases	8 (17.39%)	4 (10.81%)	0.369

**Table 3 diagnostics-13-03032-t003:** Comparative analysis of medical characteristics and outcomes in Group 1 and Group 2.

	Group 1 (*n* = 46)	Group 2 (*n* = 37)	*p* Value
Urban area	12 (26.08%)	18 (48.64%)	0.024
Rural area	34 (73.91%)	19 (51.35%)	0.0245
Cesarean section Delivery	16 (34.78%)	34 (91.89%)	<0.001
Normal Delivery	30 (65.21%)	3 (8.10%)	<0.001
Smoking patients	0	26 (70.27%)	<0.001
Non smoking patients	46 (100%)	11 (21.72%)	<0.001
Normal ECG	46 (100%)	24 (64.86%)	<0.001
Pathological ECG	0	13 (35.13%)	<0.001
Oxigenotherapy	17 (36.94%)	31 (83.78%)	<0.001
Airvo	4 (8.69%)	7 (18.91%)	0.148
Tracheal intubation	3 (6.52%)	6 (16.21%)	0.134
Mortality	3 (6.52%)	4 (10.81%)	0.458
More than 10 days of hospitalization	24 (52.17%)	35 (94.59%)	<0.001

**Table 4 diagnostics-13-03032-t004:** Prevalence of Pathological ECG Findings in Patients with SARS-CoV-2 Infection.

Pathological ECG	13 (15.66%)	Definition
Ventricular extrasystoles	3 (3.61)	Ventricular extrasystoles result from premature excitation of the heart from a site beyond the bifurcation of the bundle of His, at the level of the conductive tissue or myocardial cell [9]
Sinus Bradycardia	4 (4.81)	Bradycardia is a commonly observed arrhythmia and a frequent occasion for cardiac consultation. Defined as of less than 50–60 bpm [10]
Sinus Tachycardia	2 (2.4%)	Sinus tachycardia is a consistent heart rhythm characterized by a faster-than-normal heart rate, leading to an elevated cardiac output [11]
Long QT Syndrome	2 (2.4%)	The QT interval observed on an electrocardiogram (ECG) signifies the length of the ventricular action potential, which corresponds physiologically to the duration of ventricular depolarization and repolarization [12]
Larger T wave	2 (2.4%)	Tall T waves can also be signs of ischemic changes and hyperkalemia. Additionally, T waves may be tall as a normal variant [13]

**Table 5 diagnostics-13-03032-t005:** Distribution of SARS-CoV-2 symptomatology at admission between Group 1 and Group 2.

Symptoms	Group 1 (*n* = 46)	Group 2 (*n* = 37)	*p* Value
Ageusia	16 (34.78%)	8 (21.62%)	0.164
Anosmia	14 (30.43%)	17 (45.94%)	0.124
Cough	40 (86.95%)	35 (94.59%)	0.214
Fever	44 (95.65%)	19 (51.35%)	<0.001
Dyspnea	37 (80.43%)	33 (89.18%)	0.246
Rhinorrhea	31 (67.39%)	16 (43.24%)	0.246
Fatigability	43 (93.47%)	31 (83.78%)	0.0196
Asymptomatic	10 (21.73%)	2 (5.40%)	0.0262

**Table 6 diagnostics-13-03032-t006:** Use of medications in patients with SARS-CoV-2 infection.

Medication	Group 1 (*n* = 46)	Group 2 (*n* = 37)	*p* Value
Low molecular mass anticoagulant	36 (78.26%)	37 (100%)	0.0013
Corticoid therapy	15 (32.60%)	33 (89.18%)	<0.001
Vitamins	46 (100%)	35 (94.59%)	0.0475
Antibiotics	27 (58.69%)	29 (78.37%)	0.0434
Antalgic	29 (63.04%)	30 (81.08%)	0.0562
Non-steroidal anti-inflammatory drugs	18 (39.13%)	28 (75.67%)	<0.001
Antivirals	9 (19.56%)	30 (81.08%)	<0.001

## Data Availability

The datasets used and/or analyzed during the current study are available from the first author.

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
