# Peer review of "NT-proBNP—Possible Prognostic Marker in Pregnant Patients with Associated Cardiovascular Risk Factors and SARS-CoV-2 Infection"

_diagnostics, 2023, doi:10.3390/diagnostics13193032_

Round 1
Reviewer 1 Report
The authors describe the additional use of the biomarker NT-proBNP to identify patients at risk for an adverse outcome and having SARS-Cov-2 infection. An increase of NT-proBNP was especially pronounced in the cohort with cardiovascular risk factors ans COVID-19. Although the methods and tables and figures are adquate, the authors should comment on the fact that patients with type 1 and type 2 diabetes were included into groupe 1 which should be without cardiovascular risk factors?
Maybe the authors should as well include the citation of the ESC guideline for the management of cardiovascular diseases during pregnancy (PMID: 30165544) as NT-proBNP was stressed in this manuscript as an important predictor of adverse events. Further, the current ESC heart failure guidelines should be cited instead of the old 2016 version (PMID: 34447992)
The English language used in the manuscript is sound and only needs minor corrections as far as a non-native English speaker can assess this aspect.
Author Response
Thank you for your constructive feedback on our manuscript. We appreciate the recognition of the insights we have provided NT-proBNP as a possible prognostic marker in pregnant patients with associated cardiovascular risk factors and SARS-COV-2 infection.
We are committed to addressing and rectifying the issues you have pointed out. Your detailed review will undoubtedly enhance the quality and clarity of our research. We will ensure that each point is thoroughly considered and implemented as necessary in the revised version of our manuscript.
- The authors describe the additional use of the biomarker NT-proBNP to identify patients at risk for an adverse outcome and having SARS-Cov-2 infection. An increase of NT-proBNP was especially pronounced in the cohort with cardiovascular risk factors ans COVID-19. Although the methods and tables and figures are adquate, the authors should comment on the fact that patients with type 1 and type 2 diabetes were included into groupe 1 which should be without cardiovascular risk factors?
Thank you for your feedback on our tables and figures. Type 1 and type 2 diabetes were included in group 1 because of an omission and a mistake. We are sorry for the mistake and we clarified this, now, in our article. We appreciate your guidance on this matter.
- Maybe the authors should as well include the citation of the ESC guideline for the management of cardiovascular diseases during pregnancy (PMID: 30165544) as NT-proBNP was stressed in this manuscript as an important predictor of adverse events. Further, the current ESC heart failure guidelines should be cited instead of the old 2016 version (PMID: 34447992)
We have followed your instruction and included the ESC guideline for the management of cardiovascular diseases during pregnancy (PMID: 30165544), also instead of the old 2016 version (PMID: 34447992) we cited the current ESC heart failure guidelines.
- The English language used in the manuscript is sound and only needs minor corrections as far as a non-native English speaker can assess this aspect.
We revised our article and rectified the English language mistakes as best as we could.
Once again, we thank you for your time and effort in reviewing our work. Your input is greatly
appreciated, and we look forward to enhancing the manuscript based on your valuable insights.
Best regards,
C.-I. Marta
-----------------------------------------------------
Reviewer 2 Report
The subject is interesting and useful in the context of COVID19 pandemic and the approach of this topic is meritorious.
The topic addressed is promising as possible results that can be obtained and I believe that the collected data could be better used through an adequate statistical processing that could increase significantly the value of the manuscript.
The tables in the article strictly show the percentage distribution between the two groups of patients (with and without cardiovascular risk factors). There is only one statistical test mentioned, just one p-value mentioned and I would advise you to seek the help of a statistician.
It would be interesting to follow the correlations between the NT-proBNP value and the degree of severity of the infectious disease, the degree of lung damage, the need for oxygen, the number of days of hospitalization, the number of weeks of pregnancy or different correlations with the associated pathologies.
Also, please discuss with a cardiologist. The T wave represent the repolarization of the ventricles and when it is tall and large can be a sign of cardiac ischemia instead of cardiac hypertrophy. The QRS complex is more suitable for assessing ventricular hypertrophy.
Author Response
Thank you very much for taking the time to review our manuscript titled NT-proBNP – possible prognostic marker in pregnant patients with associated cardiovascular risk factors and SARS-COV-2 infection. We sincerely appreciate your feedback and your kind words regarding the importance of the topic. We are committed to improving the manuscript and addressing the issues you have pointed out.Your feedback is invaluable in this process, and we will carefully review and revise the manuscript to ensure its quality and comprehensibility.
- The tables in the article strictly show the percentage distribution between the two groups of patients (with and without cardiovascular risk factors). There is only one statistical test mentioned, just one p-value mentioned and I would advise you to seek the help of a statistician.
In response to your suggestion regarding the statistical tests, we have taken the initiative to improve the tables and to expand our statistical analysis on the studied groups.
- It would be interesting to follow the correlations between the NT-proBNP value and the degree of severity of the infectious disease, the degree of lung damage, the need for oxygen, the number of days of hospitalization, the number of weeks of pregnancy or different correlations with the associated pathologies.
In response to your suggestion, we intend to conduct a new study that tracks specifically these correlations throughout pregnancy and COVID-19 viral infections which will include a larger cohort throughout a longer period of time.
- Also, please discuss with a cardiologist. The T wave represent the repolarization of the ventricles and when it is tall and large can be a sign of cardiac ischemia instead of cardiac hypertrophy. The QRS complex is more suitable for assessing ventricular hypertrophy
In accordance with your suggestion, we have adjusted the explanation of our table and modified the confusion and mistake we have made. We appreciate your guidance on this matter and have made the necessary adjustments in the article.
We thank you for your time and effort in reviewing our work. We look forward to enhancing the manuscript based on your valuable insights.
Best regards,
C.-I. Marta
=================================
Round 2
Reviewer 2 Report
The authors addressed the major issues that were raised.
Author Response
We thank you again for your constructive feedback on our manuscript, and for your time and effort in reviewing our work.
Best regards,
C.-I. Marta